# Sex Differences in a Novel Mouse Model of Spinocerebellar Ataxia Type 1 (SCA1)

**DOI:** 10.3390/ijms26062623

**Published:** 2025-03-14

**Authors:** Adem Selimovic, Kaelin Sbrocco, Gourango Talukdar, Adri McCall, Stephen Gilliat, Ying Zhang, Marija Cvetanovic

**Affiliations:** 1Department of Neuroscience, University of Minnesota, 2101 6th Street SE, Minneapolis, MN 55455, USA; selim023@umn.edu (A.S.); taluk012@umn.edu (G.T.); mccal221@umn.edu (A.M.); zhan2142@umn.edu (Y.Z.); 2Graduate Medical Sciences-Anatomy & Neurobiology, Boston University, Boston, MA 02118, USA; ksbrocco@bu.edu; 3Department of Pharmacology, Yale School of Medicine, Yale University, New Haven, CT 06511, USA; stephen.gilliat@yale.edu; 4Institute for Translational Neuroscience, University of Minnesota, 2101 6th Street SE, Minneapolis, MN 55455, USA

**Keywords:** SCA1, neurodegeneration, motor deficits, cognitive deficits, Purkinje cell pathology, astrocytes, microglia, transcriptomics

## Abstract

Spinocerebellar ataxia type 1 (SCA1) is a rare autosomal dominant inherited neurodegenerative disease caused by the expansion of glutamine (Q)-encoding CAG repeats in the gene *ATAXIN1* (*ATXN1*). Patients with SCA1 suffer from movement and cognitive deficits and severe cerebellar pathology. Previous studies identified sex differences in disease progression in SCA1 patients, but whether these differences are present in mouse models is unclear. Using a battery of behavioral tests, immunohistochemistry of brain slices, and RNA sequencing, we examined sex differences in motor and cognitive performance, cerebellar pathology, and cerebellar gene expression changes in a recently created conditional knock-in mouse model *f-ATXN1^146Q^* expressing human coding regions of *ATXN1* with 146 CAG repeats. We found worse motor performance and weight loss accompanied by increased microglial activation and an increase in immune viral response pathways in male *f-ATXN1^146Q^* mice.

## 1. Introduction

Numerous studies have highlighted differences between sexes in terms of the prevalence, progression, and outcomes of various major neurodegenerative diseases [1,2,3,4,5,6,7]. Spinocerebellar ataxia type 1 (SCA1) is an autosomal dominant inherited neurodegenerative disease caused by the abnormal expansion of glutamine (Q)-encoding CAG repeats in the coding region of the *Ataxin-1* (*ATXN1*) gene [8,9]. SCA1 patients suffer from progressive motor deficits and profound cerebellar pathology [5]. SCA1 belongs to the group of polyglutamine diseases including Huntington’s disease, Dentatorubral–pallidoluysian atrophy (DRPLA), spinal and bulbar muscular atrophy (SBMA), SCA 1, 2, 3, 6, 7, and SCA 17 [10,11]. This diverse group of neurodegenerative disorders is characterized by progressive motor impairments impacting coordination, balance, and speech [10]. As these diseases are rare, there are limited studies examining whether sex impacts the clinical symptoms and the progression of ataxias. One notable feature of many of these polyglutamine diseases is a phenomenon where the number of CAG repeats expands or contracts in each generation. Previous studies identified that the sex of the parent influences CAG expansion in the offspring, as children inherited elongated CAGs when the father was affected [4]. In addition to inheritance, sex was reported to influence disease progression as clinical studies demonstrated faster disease progression in female HD and SCA patients [4], especially with motor difficulties. For example, female SCA1, SCA3, and SCA6 patients experience more severe motor and cognitive impairments, and an earlier age of onset compared to male patients [5,6,7].

Despite the importance of understanding the sex differences for therapeutic approaches, little is known about the biological mechanisms underlying the accelerating progression in female SCA and HD patients.

Much of our understanding of the underlying mechanisms of SCA1 pathogenesis comes from research using mouse models [12]. We previously compared sex differences in both the SCA1 transgenic mouse model, *ATXN1[82Q]*—overexpressing human *ATXN1[82Q]* 30-60X selectively in Purkinje cells—and the SCA1 knock-in model, *Atxn1^154Q/2Q^*—expressing mouse *Atxn1[154Q]* throughout the brain at endogenous levels. We demonstrated worse motor and cognitive performance in male *ATXN1[82Q]* mice, but we found no statistically significant sex differences in the behavioral phenotype in *Atxn1^154Q/2Q^* mice. This raised two important questions: whether these sex differences seen in the transgenic model are due to the presence of human ATXN1 (instead of the mouse ATXN1 found in the knock-in model) or to exacerbated cerebellar pathology due to the overexpression of ATXN1 by 30–60-fold in transgenic SCA1 mice. To determine the im-pact of human vs. mouse mutant ATXN1, as wells as Purkinje-cell-specific vs. brain-wide expression of mutant ATXN1 on sex differences in SCA1 models, here we use the recently created, first knock-in mouse model of SCA1 expressing expanded human ATXN1 with 146Q under the endogenous mouse *Atxn1* promoter, *f-ATXN1^146Q^* line [13]. Like human patients, these mice express one unexpanded and one expanded ATXN1 at a physiological spatiotemporal pattern. We have previously reported that *f-ATXN1^146Q^* mice exhibit impaired performance on the rotarod and balance beam motor tests, demonstrating that they model SCA1-like large and finer motor deficits [13]. In addition, to investigate whether sex impacts SCA1 progression, we compared male and female mice at two different stages of disease. Furthermore, to increase mechanistic understanding of sex differences, we analyzed cellular and molecular pathogenesis in female and male SCA1 mice. This analysis aims to inform the development of effective treatments that consider the influence of sex on the progression and manifestation of this neurodegenerative disorder.

## 2. Results

### 2.1. Male SCA1 Mice Perform Worse on the Rotarod and Show Weight Loss

To characterize sex differences in SCA1 mice throughout disease progression, we used a battery of tests at different ages and then compared the cellular and molecular changes in these mice (Figure 1A).

Rotarod is a four-day-long gold-standard test in which mice are made to run on the accelerating rotating rod. It is used to evaluate gross motor and coordination deficits in mouse models of disease [14]. A decrease in the time to stay on the rod, termed latency to fall, indicates motor and balance impairment. To capture the progressive nature of motor and balance impairments in SCA1 mice, we compared male and female SCA1 mice to the sex matched wild-type (WT) control mice, on the rotarod at three different ages: 12–14 weeks of age, when motor deficits become first detectable; 22–26 weeks, when mice start exhibiting muscle deficits and weight loss, and 33–35 weeks, right before the final stage when mice need to be euthanized to prevent suffering [13].

Male *f-ATXN1^146Q^* mice performed significantly worse compared to their WT male controls at each day and age they were tested (Figure 1B). Compared to WT female controls, female *f-ATXN1^146Q^* mice were significantly impaired on day 4 at 12–14 weeks, day 1 at 22–26 weeks, and on all four days at 33–35 weeks of age. In addition, female *f-ATXN1^146Q^* mice performed significantly better compared to male *f-ATXN1^146Q^* mice at day 4 at 22–26 weeks and on all four days at 33–35 weeks.

The balance beam test is used to evaluate fine movement and coordination impairments by quantifying the amount of time it takes for mice to cross the beam (latency to cross). We found that male mice had significantly longer latency to cross compared to their WT male controls at 22 and 33 weeks of age (Figure 1C). This slower movement on the beam may indicate balance and movement impairment. Female *f-ATXN1^146Q^* mice showed a trending reduction in latency to cross at both ages that did not reach statistical significance. Together, these results may indicate worse motor deficits in male compared to female *f-ATXN1^146Q^* mice.

### 2.2. Sex Differences in Terms of Failure to Gain Weight and Weight Loss in f-ATXN1^146Q^ Mice

Wild-type mice gain weight throughout life (Figure 1D). We found that both female and male *f-ATXN1^146Q^* mice do not increase in weight after 12–14 weeks, indicating a ‘failure to gain weight’ phenotype. However, while female *f-ATXN1^146Q^* mice maintain a constant weight, male *f-ATXN1^146Q^* start losing weight after 14 weeks, indicating a ‘loss of weight’ phenotype.

### 2.3. Sex Differences in Cognitive Deficits in f-ATXN1^146Q^ Mice

To evaluate the progression of cognitive deficits, mice were trained and tested in the Barnes maze at 12 weeks of age. During the four training days, mice are placed on a circular table with holes around the perimeter [15]. Only one of the holes (escape hole) leads to a box, and mice use spatial cues to reach the escape hole as fast as they can (latency to escape). With each day of training, all mice have progressively shorter latency to escape (Figure 2A), indicating an ability to learn. However, male *f-ATXN1^146Q^* were significantly slower to escape compared to the wild-type males on the last two days of training. After four days of training, mice undergo a probe test, where all the holes are covered, and memory is evaluated as the increased time mice spend in the quadrant where the escape hole was previously. Both female and male mice were impaired on the test day at 12 weeks of age (Figure 2B).

At 24 weeks, we evaluated long-term memory by re-introducing mice to the Barnes maze and measuring time in the goal zone during a probe test. While both male and female *f-ATXN1^146Q^* mice exhibited lower goal zone times, indicative of long-term memory deficits, this only reached statistical significance for male mice (Figure 2C).

To evaluate learning and memory at this later stage of disease, we retrained mice for three days with a different location of the escape hole. While both male and female *f-ATXN1^146Q^* mice were slower in finding the escape hole compared to sex-matched wild-type mice, with every day of training they were still faster in finding the escape hole, indicating that they are still able to acquire new memories (Figure 2D). However, the following day, female mice spent significantly less time in the new goal zone compared to both their female WT controls and male *f-ATXN1^146Q^* mice (Figure 2E).

We also tested memory using contextual fear conditioning. In contextual fear conditioning, mice undergo one day of training in which five electric shocks are delivered in the context of a specific smell and chamber floor texture (context 1). Wild-type mice exhibited a fear response measured as increasing freezing time after each shock (Figure 2F). While female *f-ATXN1^146Q^* mice exhibited similar freezing responses to female wild-type mice, male *f-ATXN1^146Q^* mice were freezing significantly less compared to their wild-type male controls. The following day, mice were tested for associative fear memory by measuring freezing in the context 1 chamber, but without any shocks. Female *f-ATXN1^146Q^* mice exhibited similar freezing to female wild-type mice, and male *f-ATXN1^146Q^* mice were freezing significantly less compared to their wild-type male controls (Figure 2G).

We then re-introduced mice to the context 1 chamber at 25 weeks to test how much of their fear response remained after 12 weeks. As expected, all mice were freezing less after 12 weeks, indicating forgetting, but there was a trend towards more forgetting in male mice compared to female mice. As such, freezing responses of wild-type and *f-ATXN1^146Q^* male mice were not different. On the other hand, female *f-ATXN1^146Q^* mice exhibited an increased amount of forgetting and were freezing much less (Figure 2H). Together, these results may suggest worse cognitive impairments in female compared to male *f-ATXN1^146Q^* mice.

We also tested whether fear-associative learning and memory become worse with disease progression. Mice were re-trained in a different context for one day, and all mice exhibited a strong freezing response that was slightly but significantly decreased in both male and female mice (Figure 2I). We did not detect any differences in the time spent freezing during testing the next day (Figure 2J). This result indicates that mice develop a strong fear response with aging that may mask any deficits in *f-ATXN1^146Q^* mice.

### 2.4. Sex Differences in Cerebellar Pathology

To measure cerebellar pathology, we evaluated atrophy/shrinking of PC dendrites as a reduced width of the molecular layer and intensity of calbindin staining [16,17,18]. Using PC-specific marker calbindin (Figure 3A), we found a similarly reduced width of the molecular layer (Figure 3B) and intensity of calbindin staining in both male and female *f-ATXN1^146Q^* mice, indicating similar levels of PC dendritic atrophy (Figure 3C).

Using an increase in intensity of GFAP (Figure 3D) as a general marker of reactive astrocytes, which we have previously used to demonstrate astrogliosis in SCA1 mice [19,20,21], was found to significantly increase GFAP in both female and male *f-ATXN1^146Q^* mice compared to sex-matched WT controls. We also found significantly increased relative GFAP intensity in female compared to male *f-ATXN1^146Q^* mice (Figure 3D,E). Using Iba1 staining to quantify microglia (Figure 3F), we found that only male *f-ATXN1^146Q^* mice had significant increases in microglia density compared to sex-matched WT controls (Figure 3G), indicating sex differences in glial pathology in SCA1 mice.

To evaluate molecular differences in SCA1, we performed RNA sequencing on cerebellar extracts to determine if transcriptional perturbations caused by expanded ATXN1, thought to be one of the key mechanisms of SCA1 pathogenesis [14,22], exhibit any sex differences in *f-ATXN1^146Q^* mice. Compared to the female WT mice, we found 852 DEGs in female *f-ATXN1^146Q^* mice, out of which 418 (approximately half) were downregulated, and 434 DEGs were upregulated (Figure 4A). In contrast, compared to male WT mice, we found 524 DEGs in male *f-ATXN1^146Q^* mice, with 351 DEGs (67%) downregulated and 173 DEGs upregulated (Figure 4B). To investigate regulatory pathways involved in SCA1 transcriptional changes, we performed transcription factor analysis (Figure 4C). Among the three top transcriptional regulators of downregulated DEGs in female *f-ATXN1^146Q^* mice was CTCF transcriptional regulator—known to interact with estrogen receptor [23]—and two different Kruppel-like (KLF) factors—well known for their role in disease and inflammation [24]. Interferon-regulatory factors (IRF) 3 and 9 were the top transcriptional regulators of upregulated DEGs in female *f-ATXN1^146Q^* mice. Wilm’s tumor 1 (WT1), associated with Alzheimer’s disease and synaptic plasticity [25], is among the key regulators of downregulated DEGSs, while heat shock response factor 4 (HSF 4), a negative regulator of HSFs [26] and E24F a transcription factor known to be involved in Alzheimer’s disease pathogenesis [27], were identified as a key regulators of upregulated DEGs in male *f-ATXN1^146Q^* mice.

To understand the functional implications of both upregulated and downregulated gene expression, we performed GO (Figure 4D) and KEGG (Figure 4E) pathway analyses. Among the pathways impacted by upregulated DEGs were calcium and ion binding in females, and viral infections in male *f-ATXN1^146Q^* mice. Neuroactive ligand–receptor interaction and aldosterone synthesis and secretion were impacted by downregulated DEGs in female *f-ATXN1^146Q^* mice, and the extracellular matrix was identified as a pathway that was impacted in male *f-ATXN1^146Q^* mice.

We found large and variable perturbations in the expression of estrogen-regulated genes in the cerebella of female and male *f-ATXN1^146Q^* mice compared to their wild-type controls (Figure 4F, Appendix A). Interestingly, certain groups of genes such as *Ddx5 Uba5*, and *Rbfox2* were strongly upregulated in male *f-ATXN1^146Q^* mice but decreased in female *f-ATXN1^146Q^* mice. *Ddx5* encodes the DEAD-box RNA helicase DDX5 (also known as p68) inhibitor of anti-viral immune response, and acts as the regulator of complement 3 (C3) and myelin basic protein (MBP) expression [28,29]. Mutations in *UBA5* that encode ubiquitin-like modifier-activating enzyme 5 (UBA5) are the cause of autosomal recessive spinocerebellar ataxia-24 (SCAR24) [30]; moreover, splicing regulator Rbfox2 is required for Purkinje cell function [31]. *Estrogen receptor 1 (Esr1)* is upregulated in female *f-ATXN1^146Q^* mice.

## 3. Discussion

Our previous studies revealed sex differences in cognitive and motor deficits in Purkinje-cell-specific transgenic SCA1 model, *ATXN1[82Q]* line, overexpressing human mutant ATXN1 but not in the SCA1 knock-in model, *Atxn1^154Q/2Q^* line expressing mouse ATXN1 under endogenous promoter. This raised two important questions: whether these sex differences seen in the transgenic model are due to human vs. mouse ATXN1 or to exacerbated cerebellar pathology due to the overexpression of ATXN1 by 30–60-fold in transgenic SCA1 mice. To address these questions, here we examined sex differences in the motor and cognitive behavior, in the novel mouse model of SCA1, *f-ATXN1^146Q^* mice, expressing human mutant ATXN1 under endogenous promoter. Overall, we found that male *f-ATXN1^146Q^* mice are more impacted. Using rotarod and balance beam we found earlier and more severe motor deficits in male *f-ATXN1^146Q^* mice at all ages tested. This contrasts with earlier onset of motor deficits in female *ATXN1[82Q]* mice, but worse deficits in male *ATXN1[82Q]* mice at later disease stage. Cognitive performance was more complex. Similarly to our study in *Atxn1^154Q/2Q^* mice, both male and female *f-ATXN1^146Q^* mice were similarly impaired in learning and memory in the Barnes maze at 12 weeks, indicating the absence of sex differences in cognitive deficits early in disease. However, at 24 weeks, female *f-ATXN1^146Q^* mice performed worse in the Barnes maze, indicating that with disease progression, female SCA1 mice became more cognitively impaired, similar to human SCA patients.

We found that only male *f-ATXN1^146Q^* mice were freezing significantly less in the contextual fear conditioning both during training and testing at 12–13 weeks, indicating the deficits in learned fear response. At 24–25 weeks, neither male nor female *f-ATXN1^146Q^* mice were impaired compared to their wild-type controls, indicating that with aging, SCA1 mice normalize freezing in response to noxious stimuli.

Little is understood about ‘failure to gain’ or ‘lose weight’ phenotypes in SCA1. Surprisingly, while male *f-ATXN1^146Q^* mice progressed from failure to gain weight at 12–13 weeks to the ‘loss of weight’ phenotype at 33–35 weeks, female *f-ATXN1^146Q^* mice exhibited only failure to gain weight at all ages examined, indicating sex differences in the weight phenotype in SCA1.

Sex differences in cerebellar pathology of SCA1 mouse models have not been previously investigated. Dendritic atrophy and decreases in calbindin staining are often used to evaluate PC pathology in SCA1 mice. We have found no differences in dendritic atrophy and calbindin intensity in male and female *f-ATXN1^146Q^* mice. On the other hand, our results show slightly increased astrogliosis in female and microgliosis in male *f-ATXN1^146Q^* mice, respectively, indicating, for the first time, sex differences in cellular SCA1 pathogenesis. While slight, it is possible that sex differences in glial neuroinflammatory phenotypes underly sex differences in SCA1 behavior. Future studies using samples from SCA1 patients matched for repeat number and the length of disease could be used to determine whether these sex differences in glial activation exist in SCA1 patients.

We also observed differences in molecular pathogenesis in female and male SCA1 mice. In general, transcriptional changes were more abundant in female SCA1 mice, indicating, for the first time, sex differences in transcriptional changes in SCA. We also found that male *f-ATXN1^146Q^* mice had a pronounced increase in viral response pathways, consistent with increased microglial density.

Our results indicate a potentially worse motor phenotype in male *f-ATXN1^146Q^* mice throughout disease progression, and worse cognitive deficits in female *f-ATXN1^146Q^* mice at later disease stages. Studies in patients indicate worse cognitive progression in female SCA patients, consistent with worse cognitive phenotypes and increased transcriptional perturbations in female SCA1 mice. On the other hand, we have previously found that sex affects ataxia severity in SCA1 after adjusting for age, disease duration, and CAG repeats, with women with SCA1 having SARA scores that are 3.32 points higher than men with SCA1 [7]. While the reasons for these species differences in motor phenotype remain unclear at this point, hormonal status is one potential contributing factor. The onset of SCA1 is usually in the age range of late thirties to early forties; thus, most SCA1 female patients will experience menopause and decreased protective effects of estrogen [32]. In age-dependent movement disorders, female patients may be more impacted, with disease severity fluctuating during the menstrual cycle, and indications of a potential worsening of the disease phenotype in postmenopausal women [2,3,33].

While mice do not experience menopause like humans, they undergo a loss of ovarian function around 12 months of age [34]. As SCA1 mice tend to die prematurely before 12 months of age [13], female *f-ATXN1^146Q^* mice continue to be protected by estrogen. Consistent with this, we have found differences in the expression of estrogen-regulated genes in SCA1 mice, including increased expression of *Esr1* in female *f-ATXN1^146Q^* mice. Future studies could manipulate estrogen levels in female mice to directly examine the role of estrogen signaling in SCA1 pathogenesis.

## 4. Materials and Methods

### 4.1. Mice

Mice were housed in a temperature- and humidity-controlled room on a 12 h light/12 h dark cycle with access to food and water ad libitum. Mice were tested starting from 12 weeks of age. All animal experiments were performed in compliance with the National Institutes of Health’s Guide for the Care and Use of Laboratory Animals and approved by the University of Minnesota Institutional Animal Care and Use Committee. In all experiments, littermate wild-type controls were used when possible. The *f-ATXN^146Q^* mice [13] were gifts from Dr. Harry Orr and Michael Koob.

### 4.2. Genotyping

PCR was performed with the following primers (Integrated DNA Technologies, Coralville, IA, USA) to determine which animals have an expanded *f-ATXN1^146Q^* allele: ATXN1-146Q repeat forward (5′-CAACATGGGCAGTCTGAG-3′) and ATXN1-146Q repeat reverse (5′-GTGTGTGGGATCATCGTCTG-3′).

### 4.3. Behavioral Testing

Sample sizes in the behavioral tests were determined using power analysis and prior experience with these tests, or previous reports using similar methodology. Experimenters were blinded to the genotype during all tests.

### 4.4. Motor Testing

Rotarod A rotarod apparatus (Ugo Basile, Gemonio, Italy) was used to assess motor coordination and balance. Testing occurred over four consecutive days, and each test day consisted of 4 individual trials (~15 min intertrial interval) where mice were placed on a rotating rod that accelerated from 5 to 40 RPM in 1 RPM steps over the course of a 5 min interval (~7 s ramp). The trial ended when mice fell from the rod, when they failed to continuously walk on the rod (held on to the rod instead of walking for two full 360° rotations), or after the 5 min maximum trial time was reached. The apparatus was cleaned with 70% ethanol or a hydrogen-peroxide-based cleaning solution between animals and between each trial. The latency to fall (s) was averaged across the 4 trials in individual animals for analysis.

Balance beam The beam walking assesses a mouse’s ability to maintain balance while traversing a narrow beam to reach a safe platform. The beam is made of wood, and is 17 mm in diameter, 1 m long, and elevated 50 cm above the bench by metal supports. The mouse is placed on the beam at one end and allowed to walk to the goal box (20 × 20-cm^2^, with a 4 × 5 cm^2^ entrance hole). The recorded measurements include the time taken to cross the beam and the number of paw faults or slips. The entire test takes approximately 60 s.

### 4.5. Cognitive Testing

Barnes maze. The maze was a white circular platform 91 cm in diameter with twenty 5 cm circular holes spaced evenly around the edge, raised approximately 92 cm above the floor. One of the holes led to a 5 cm wide × 11 cm long × 5 cm deep opaque box (the “escape box”), and the other nineteen were covered. The testing room had visual cues on the walls to serve as landmarks, and all objects in the room were kept in the same places for every trial. The position of each mouse was tracked using AnyMaze software (7.49). Mice were exposed to the maze for four 3 min trials per day over four consecutive training days (intertrial interval of approximately 15 min). Mice who did not enter the escape box within 3 min were gently guided to it. Training day data are reported as a path length (distance traveled before entering the escape hole) and analyzed by two-way repeated measures ANOVA. A probe test was conducted 24 h after the last training session. For the probe test, the escape hole was covered, and each mouse was allowed to explore the maze freely for 90 s. The time spent in each quadrant of the maze was recorded, and the amount of time spent in the goal quadrant (the quadrant centered on the goal hole) was analyzed by one-way ANOVA.

Contextual fear conditioning. Conditioning took place in chambers with a floor consisting of stainless steel rods through which shocks were delivered (Med Associates #ENV-008-FPU-M, St. Albans, VT, USA). On day 1, mice were placed in the chambers for a 10 min period, during which they received five shocks to the feet (0.70 mA, 2 s duration). Freezing for 60 s after each shock was quantified automatically using VideoFreeze software version 1.0 (freezing was defined as a motion index ≤15 lasting ≥500 ms). Next, 24 h after the initial conditioning, mice were returned to the same chambers with the shock generators turned off, and freezing behavior was monitored for 3 min. For 1–2 h after being placed in the conditioned context, mice were placed in a second context for 3 min to measure baseline freezing. The baseline context used the same chambers but differed from the conditioned context in terms of floor texture (smooth plastic versus metal rods), shape (curved plastic wall versus square metal wall), and odor (0.5% vanilla extract versus 33% Simple Green). The acquisition of freezing responses is reported as the percentage of time that mice are freezing in the 60 s period following each of the 5 shocks, analyzed by two-way repeated measures ANOVA. A 24 h recall is reported as the percentage that is freezing in each context over the 3 min test period, analyzed by two-way repeated measures ANOVA.

### 4.6. RNAseq

RNA was extracted from dissected cerebella using TRIzol Reagent (Life Technologies, Carlsbad, CA, USA). RNA was sent to the University of Minnesota Genomics Center for quality control, including fluorometric quantification (RiboGreen assay, Life Technologies) and RNA integrity with capillary electrophoresis (Agilent BioAnalyzer 2100, Agilent Technologies Inc., Santa Clara, CA, USA). All samples with RNA integrity numbers (RINs) 8.0 or greater proceeded to RNA sequencing on an Illumina NextSeq 550 using a 100 nt paired-end read strategy [29]. Data are stored on University of Minnesota Supercomputing Institute Servers. To analyze the data, raw, paired short reads were analyzed through CHURP (Collection of Hierarchical UMII-RIS Pipelines, PEARC ’19 Proceeding, Article No. 96), which include data quality control via FastQC (v0.12.1), data preprocessing via Trimmomatic, mapping via HiSat2 (v2.1.0) against the reference mouse genome, and expression quantification via Subread (v2.0.6). The resulting count matrix of gene expression was used as input to R (https://www.r-project.org/ accessed on 12 May 2024) package DESeq2 (v1.44.0), which tested the differential gene expression, visualized the behavior of expected invariant control genes, and published positive control genes. Genes with resulting raw testing *p* values less than 0.05 and an absolute log-transformed fold change greater than 0.6 were collected for further exploration. DESeq2 was used to reveal sample outliers through Principal Component Analysis (PCA). Pathway analysis and dot plot creation were preformed using the clusterProfiler package (v3.18.1) and gProfiler (PMID: 37144459) using the identified DEGs. Volcano plots were created using EnhancedVolcano (v1.8.0) packages. Genes that are involved in estrogen- regulated pathways are determined by Go Term—GO:0030520 (GOBP_INTRACELLULAR_ESTROGEN_RECEPTOR_SIGNALING_PATHWAY) and can be downloaded from MSigDB (https://www.gsea-msigdb.org/gsea/msigdb/index.jsp (accessed on 24 January 2025). Variance stabilizing transformed (DESeq2 function vst) gene expression was used as input to the R package pheatmap (v 1.0.12) to generate an expression heat map, in which the color gradient reflects the relative (row-scaled) average expression per gene, per genotype, and per sex. Accession numbers have not yet been obtained at the time of submission, and will be provided upon reviewer request.

### 4.7. Immunofluorescence

We used a minimum of six wild-type and *Atxn1^154Q/2Q^* sex-matched mice of each sex. IF was performed on a minimum of four different floating 40 μm thick brain slices from each mouse (four technical replicates per mouse per region or antibody of interest) in four different cohorts of mice derived from four litters. Experimenter performing staining, imaging, and analysis were blinded for the genotype until all quantification was completed. All mice within a cohort were stained at the same time. We used primary antibodies against Purkinje cell marker calbindin (rabbit, Sigma-Aldrich, C9848, St. Louis, MO, USA), astrocytic marker glial fibrillary acidic protein (GFAP) (chicken, Millipore, AB5541, Burlington, MA, USA), and microglial marker ionized calcium-binding adapting molecule 1 (Iba1) (rabbit, Abcam, AB107159, Cambridge, UK) as previously described [35,36,37].Confocal images of a minimum of three different slices from each mouse were acquired using a Stellaris microscope (Leica Microsystems Inc., Deerfield, IL, USA) with an oil 20× objective. Z-stacks consisting of twenty non-overlapping 1 μm thick slices were taken for each stained brain slice per brain region (i.e., four z-stacks per mouse per region, each taken from a different brain slice). The laser power and detector gain were standardized and fixed between mice within a cohort, and all images for mice within a cohort were acquired in a single imaging session to allow for quantitative comparison.

Quantitative analysis was performed using ImageJ software version 1.54j (NIH) as described previously [36]. To quantify the relative intensity of staining for calbindin, we measured the average signal intensity at three different areas in each slide and normalized it to that of the WT sex-matched control mouse of that cohort. To quantify the atrophy of the cerebellar molecular layer, we took three measurements per image of the distance from the base of the Purkinje soma to the end of their dendrites, the average being the molecular layer width for that image (min of 12 measurements per each region of interest/mouse). To quantify the relative intensity of staining for GFAP, we measured the average signal intensity in the region of interest and normalized it to that of the WT littermate mouse (minimum of 16 measurements per each mouse). We counted the number of Iba1-positive microglia in regions of interest and divided it by the size of the area to quantify microglial density.

### 4.8. Statistical Analysis

Statistical tests were performed using GraphPad Prism. 10.4.0. Data were tested for normal distribution using Kolmogrov–Smirnov and Shapiro–Wilk tests. Parametric tests were performed if normal distribution of the data was established; otherwise, non-parametric tests were chosen. Data were analyzed using two-way ANOVA or one-way ANOVA, followed by either Tukey’s, Bonferroni, or Student’s *t*-test.

## 5. Conclusions

Together, these results indicate the existence of sex differences in molecular and cellular pathogenesis, as well as in the behavioral phenotype in *f-ATXN1^146Q^* mice. As we have not seen behavioral sex differences in *Atxn1^154Q/2Q^* mice, it is possible that this is due to the expression of human mutant ATXN1 in *f-ATXN1^146Q^* mice, or due to the larger number of analyzed mice and or analysis at two different disease stages. One limitation of this study is that mouse behavior is not easily interpreted, especially in relation to human behavior. Mice are quadrupedal, and their balancing is different than in bipedal humans. However, this limitation is present in any study using mouse models. Similarly, SCA1 mice die before they lose ovarian function; thus, we cannot compare sex differences in mice at the age when most female SCA patients are more impaired. To address this limitation, future studies could use ovariectomized female SCA1 mice. Finally, species differences between human and mouse cerebellum need to be considered when interpreting our results. For instance, a recent single-nucleus RNA seq study demonstrated differences in DEGs between cerebella from SCA1 mice and patients [22]. Studies using larger numbers of samples from male and female patients could identify molecular differences in SCA1 pathogenesis. However, it is important to note that SCA1 is a rare disease, with few available samples and that the number of CAG repeats impacts the disease phenotype. Another way to address this limitation is to examine gene expression changes in cerebellar organoids that are differentiated from iPSCs derived from female and male SCA1 patients. It is also important to note that, because of the technical need to simultaneously stain and image groups, we have compared GFAP intensity between *f-ATXN1^146Q^* and WT mice of each sex, but not between male and female WT mice or female *f-ATXN1^146Q^* mice. Therefore, the relative increase in GFAP intensity in SCA1 mice compared to the sex-matched WT controls may result from lower baseline signals in WT mice and/or higher intensity in SCA1 female mice. Future studies with larger resources could address this question.

With these limitations, our study significantly increases our understanding of sex differences in SCA1 mouse models. First, we demonstrate that some sex differences, such as in motor deficits, can be observed early on and persist, with male mice being more impaired. Second, we show that other sex differences, such as cognitive deficits in the Barnes maze, are detectable only at late disease stages, with female mice being more impaired. Third, changes in glial cells and not cerebellar Purkinje neurons, show sex differences in SCA1 mice. Finally, female SCA1 mice show a larger number of transcriptional alterations, and there are surprising differences in altered pathways and transcriptional factors between female and male SCA1 mice. These results also have important implications for preclinical studies in SCA1 mice. Historically, preclinical and clinical trials and scientific research studies have been conducted predominantly on male subjects [38,39]. This male-exclusive research has resulted in significant knowledge gaps regarding potential differences in how various neurodegenerative diseases, treatments, and conditions may affect female patients. Incorporating female subjects in scientific research and preclinical studies is essential for generating comprehensive, representative, and relevant findings. In addition, failing to account for sex-based differences in research can have serious consequences as different ratios of female-to-male mice in experimental groups could lead to different outcomes irrespective of the treatment. Moreover, treatments and interventions that have been developed and approved based on studies conducted predominantly or exclusively on male subjects may not be equally effective or safe for females. Incorporating sex-specific analyses throughout the study of SCA1 is crucial for developing a more comprehensive understanding of this devastating disorder.

## Figures and Tables

**Figure 1 ijms-26-02623-f001:**
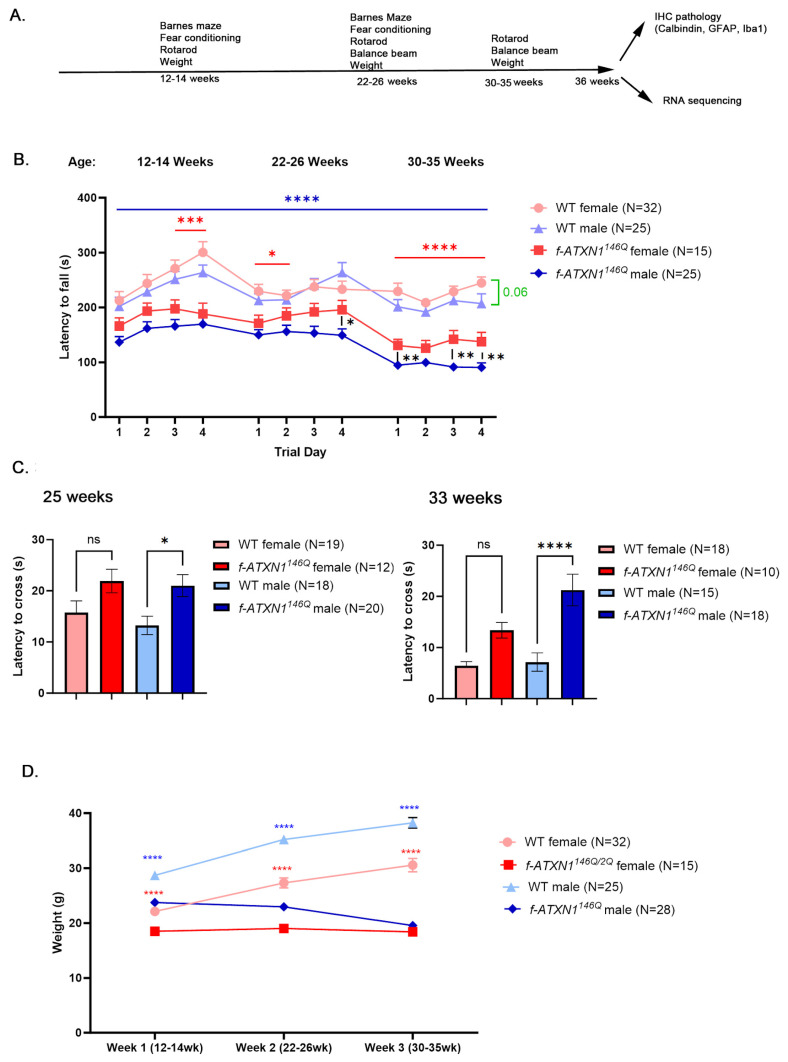
Sex differences in motor performance and ‘loss of weight’ phenotypes in SCA mice. (**A**) Study schematic. (**B**) Rotarod was used to evaluate the motor performance of male and female SCA1 and control wild-type mice at three different ages: 12–14 weeks, 22–25 weeks, and 33 weeks. (**C**) Latency to cross balance beam at 25 and 33 weeks. (**D**) Weight of mice at three different ages. Red and blue * indicate *p* values on comparison between WT and *f-ATXN1^146Q^* female and male mice, respectively. The number of mice is indicated in the legend. Data are an average ± SEM. Two-way ANOVA * *p* < 0.05, ** *p* < 0.01, *** *p* < 0.005, **** *p* < 0.001, ns means not significant.

**Figure 2 ijms-26-02623-f002:**
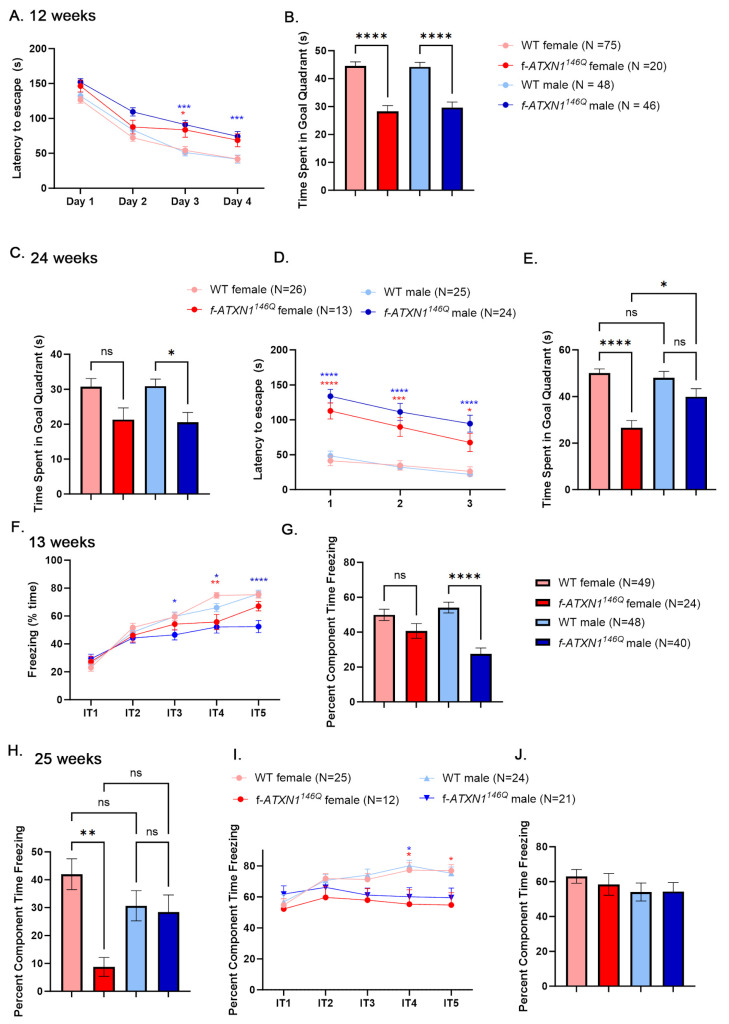
Sex differences in cognitive performance of SCA1 mice. (**A**,**B**) Barnes maze training (**A**) and testing (**B**) at 12–14 weeks. (**C**) Barnes maze re-testing at 25 weeks. (**D**,**E**) Barnes maze re-training (**D**) and testing (**E**) at 22–25 weeks. (**F**,**G**) Contextual fear conditioning (**F**) and testing (**G**) at 12–14 weeks. (**H**) Contextual fear re-testing at 25 weeks. (**I**,**J**) Contextual fear re-training (**I**) and testing (**J**) at 22–25 weeks. Data are an average ± SEM. Two-way or one-way ANOVA * *p* < 0.05, ** *p* < 0.01, *** *p* < 0.005, **** *p* < 0.001, ns means not significant. Red and blue * indicate *p* values on comparison between WT and *f-ATXN1^146Q^* female and male mice, respectively.

**Figure 3 ijms-26-02623-f003:**
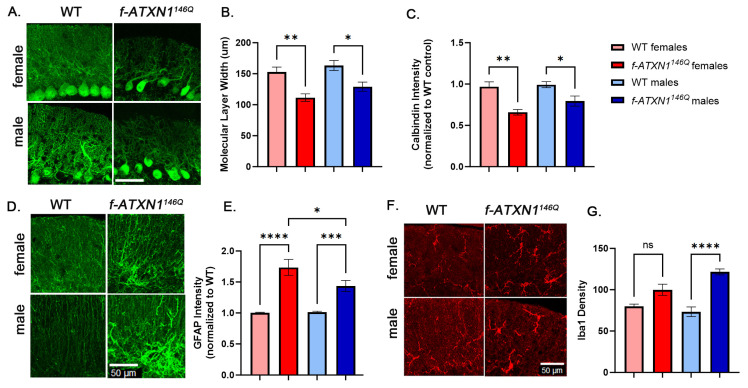
Sex differences in cerebellar pathology in SCA1 mice. (**A**) Calbindin staining was used to label Purkinje cells in 36-week-old female and male WT and *ATXN1^146Q^* mice. (**B**) Molecular layer width and (**C**) calbindin intensity were analyzed using confocal images. (**D**,**E**) GFAP staining was (**E**) quantified using confocal images from 36-week-old female and male WT and *ATXN1^146Q^* mice. (**F**,**G**) Staining with Iba1 was used to quantify microglia in the molecular layer of cerebellum by dividing the number of Iba1 microglia per area in 36-week-old female and male WT and *ATXN1^146Q^* mice (**G**). Scale bar = 50µm. Min of 3 slices from N = 6–8 mice in each condition. Data are an average ± SEM. One-way ANOVA * *p* < 0.05, ** *p* < 0.01, *** *p* < 0.005, **** *p* < 0.001, ns means not significant.

**Figure 4 ijms-26-02623-f004:**
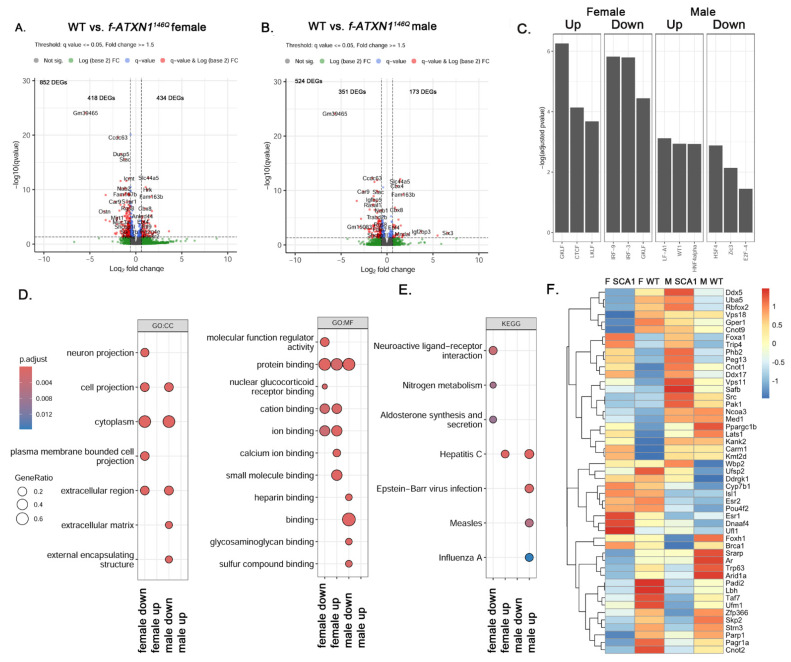
Sex differences in transcriptional changes in SCA1 cerebella. RNA was extracted from the cerebella of male and female wild-type and *ATXN1^146Q^* mice (N = 3 of each sex and each genotype) at 36 weeks. (**A**,**B**) Volcano plots of differentially expressed genes in female (**A**) and male (**B**) mice. Dash line indicates *p* < 0.05. (**C**) Transcription factor analysis of upregulated and downregulated genes in female and male *ATXN1^146Q^* mice. (**D**,**E**) GO (**D**) and KEGG (**E**) pathway analysis of up- and downregulated DEGs. (**F**) Heatmap showing the expression of estrogen-receptor-regulated genes in female and male *ATXN1^146Q^* mice.

## Data Availability

Data are available upon request.

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
