# Peer review of "Sex Differences in a Novel Mouse Model of Spinocerebellar Ataxia Type 1 (SCA1)"

_ijms, 2025, doi:10.3390/ijms26062623_

Round 1
Reviewer 1 Report (New Reviewer)
Comments and Suggestions for Authors
The authors investigated sex differences in a mouse model of Spinocerebellar Ataxia Type 1 (SCA1). The manuscript appears to be well-structured and contains a substantial amount of data. I believe the manuscript is suitable for publication in IJMS after addressing the following considerations:
(1) The authors chose to present the results separately from the discussion; however, there is no clear distinction between these sections. The results section contains numerous references, whereas it should focus solely on describing the findings of the investigation. Additionally, some sentences typical of a discussion appear in the results section, such as: "Previous studies demonstrated a failure to gain weight in a different mouse model of SCA1."
(2) In many graphs, the authors chose to use the standard error instead of the standard deviation. Please note that the most appropriate measure of data dispersion is the standard deviation rather than the standard error.
(3) The statistical analyses are reported repeatedly for each test performed. It is strongly recommended that the data analysis be described under a single subsection in the methodology to avoid redundancy.
(4) The figure legends could be more detailed. In some cases, assigning a separate letter to each graph would improve clarity and facilitate data interpretation.
Author Response
We thank the reviewer for their helpful comments and suggestions that have improved our manuscript.
The authors investigated sex differences in a mouse model of Spinocerebellar Ataxia Type 1 (SCA1). The manuscript appears to be well-structured and contains a substantial amount of data. I believe the manuscript is suitable for publication in IJMS after addressing the following considerations:
(1) The authors chose to present the results separately from the discussion; however, there is no clear distinction between these sections. The results section contains numerous references, whereas it should focus solely on describing the findings of the investigation. Additionally, some sentences typical of a discussion appear in the results section, such as: "Previous studies demonstrated a failure to gain weight in a different mouse model of SCA1."
We revised the manuscript removing these sentences from results.
(2) In many graphs, the authors chose to use the standard error instead of the standard deviation. Please note that the most appropriate measure of data dispersion is the standard deviation rather than the standard error.
We thank the reviewer for their suggestion and from now on will use SD instead of SEM. We made sure to always include information on the number of animals and that SEM was used similar to other recently published articles in IJMS, Nature and Science (Int. J. Mol. Sci. 2025, 26, 2332. https://doi.org/10.3390/ijms26052332, Nature Communication 2022, 13:753 (https://www.nature.com/articles/s41467-022-28331-7), Science Signaling, 2025, 18: 876 (https://www.science.org/doi/10.1126/scisignal.adp8973).
.
(3) The statistical analyses are reported repeatedly for each test performed. It is strongly recommended that the data analysis be described under a single subsection in the methodology to avoid redundancy.
We thank the reviewer for their suggestion. We have addressed this in each Fig legend.
(4) The figure legends could be more detailed. In some cases, assigning a separate letter to each graph would improve clarity and facilitate data interpretation.
We thank the reviewer for this suggestion. To address it we have assigned a separate letter to each graph.

Reviewer 2 Report (New Reviewer)
Comments and Suggestions for Authors
Dear Authors,
see the document below.
Additional comments:
- Confocal micrographs and graphs in the manuscript meets scientific standards and nicely and precisely shows the results, as the reference section also. No more comments about pictures, graphs and references in the manuscript.
- If I got it correct, in previous studies you investigated mouses with human version of ATXN1 gene and found differences between sexes and you also investigated mouses with mousy version of ATXN1 gene (Atxn1[154Q]) and found no differences between sexes. Please, explain more (better) why did you continue to investigate a novel mousy version of ATXN1 gene (Atxn1[146Q]) in this study or how did you get/choose this particular mouse model.
- Please explain more clearly in the discussion section with available literature about pathological sex differences obtained with the GFAP protein in Atxn1(154Q) mice, or why there is no difference between the sexes in the calbindin and Iba1 proteins, given that the latter two proteins are often used in the pathological assessment of SAC1 mice. Can the results showing slightly increased astrogliosis and microgliosis in female and male f-ATXN1146Q mice be further verified and become molecular markers for evaluation in humans with SCA1 disease?
- Please explain or how to resolve limitations that you have written in the conclusion section regarding that mouse behaviour is not easily interpreted especially in relation to human behaviour, or cellular and transcriptional cerebellar species differences between human and mice, and that SCA1 mice die before the loss of ovarian function so it’s impossible to compare sex-differences at the age when most female SCA patients are more impaired. To me this is crucial in choosing appropriate mouse model with human version of ATXN1 gene.
- This study is relevant to the field in investigating this disease, and in some ways addresses a specific gap in the field with the limitations mentioned before, but it is necessary to continue investigating and incorporating sex-specific analyses in the study of SCA1 as these results indicate existence of sex-differences in molecular and cellular pathogenesis as well as in behavioral phenotype in f-ATXN1146Q mice.

Author Response
We thank the reviewer for their helpful comments and suggestions that have improved our manuscript.
- If I got it correct, in previous studies you investigated mouses with human version of ATXN1 gene and found differences between sexes and you also investigated mouses with mousy version of ATXN1 gene (Atxn1[154Q])
and found no differences between sexes. Please, explain more (better) why did you continue to investigate a novel mousy version of ATXN1 gene (Atxn1[146Q]) in this study or how did you get/choose this particular
mouse model.
We thank the reviewer 2 for pointing this have added following to the Introduction:
“This raised two important questions: whether these sex differences seen in the transgenic model are due to the presence of human ATXN1 (instead of the mouse ATXN1 found in the knock-in model) or to exacerbated cerebellar pathology due to ATXN1 30-60 fold overexpression in transgenic SCA1 mice. To determine the impact of human vs mouse mutant ATXN1, as wells as Purkinje cell specific vs brain wide expression of mutant ATXN1 on sex differences in SCA1 models, here we use the recently created, first knock-in mouse model of SCA1 expressing expanded human ATXN1 with 146Q under the endogenous mouse Atxn1 promoter, f-ATXN1146Q line13. Like human patients, these mice express one unexpanded and one expanded ATXN1 at a physiological spatiotemporal pattern. We have previously reported that f-ATXN1146Q mice exhibit impaired performance on rotarod and balance beam motor tests, demonstrating that they model SCA1-like large and finer motor deficits13 “
- Please explain more clearly in the discussion section with available literature about pathological sex differences obtained with the GFAP protein in Atxn1(154Q) mice, or why there is no difference between the sexes in the calbindin and Iba1 proteins, given that the latter two proteins are often used in the pathological assessment of SAC1 mice. Can the results showing slightly increased astrogliosis and microgliosis in
female and male f-ATXN1146Q mice be further verified and become molecular markers for evaluation in humans with SCA1 disease?
“Sex-differences in cerebellar pathology of SCA1 mouse models have not been previously investigated. Dendritic atrophy and decrease in calbindin staining are often used to evaluate PC pathology in SCA1 mice. We have found no differences in dendritic atrophy and calbindin intensity in male and female f-ATXN1146Qmice. On the other hand, our results show slightly increased astrogliosis in female and microgliosis in male f-ATXN1146Q mice respectively, indicating, for the first time, sex-differences in cellular SCA1 pathogenesis. While slight, it is possible that sex-differences in glial neuroinflammatory phenotypes underly sex-differences in SCA1 behavior. Future studies using samples from SCA1 patients matched for repeat number and length of disease could be used to determine whether these sex differences in glial activation exist in SCA1 patients”
“It is also important to note that because of the technical need to simultaneously stain and image groups we have compared GFAP intensity between f-ATXN1146Q and WT mice of each sex, but not between male and female WT mice or female f-ATXN1146Q mice. Therefore, the relative increase in GFAP intensity in SCA1 mice compared to the sex-matched WT controls may result from lower baseline signal in WT mice and/or higher intensity in SCA1 female mice. Future studies with larger resources could address this question. “
- Please explain or how to resolve limitations that you have written in the conclusion section regarding that mouse behaviour is not easily interpreted especially in relation to human behaviour, or cellular and
transcriptional cerebellar species differences between human and mice, and that SCA1 mice die before the loss of ovarian function so it’s impossible to compare sex-differences at the age when most female SCA
patients are more impaired. To me this is crucial in choosing appropriate mouse model with human version of ATXN1 gene.
To address reviewer’s comment we added the following paragraph to the discussion:
“Mice are quadrupedal, and their balancing is different than in bipedal humans. However, this limitation is present in any study using mouse models. Similarly, SCA1 mice die before they lose ovarian function, thus we cannot compare sex-differences in mice at the age when most female SCA patients are more impaired. To address this limitation, future studies could use ovariectomized female SCA1 mice. Finally, species differences between human and mouse cerebellum need to be considered when interpreting our results. For instance, a recent single nuclei RNA seq study demonstrated differences in DEGs between cerebella from SCA1 mice and patients23. Studies using larger numbers of samples from male and female patients could identify molecular differences in SCA1 pathogenesis. However, it is important to note that SCA1 is a rare disease, with few available samples and that the number of CAG repeats impacts disease phenotype. Another way to address this limitation is to examine gene expression changes in cerebellar organoids differentiated from iPSCs derived from female and male SCA1 patients.”
- This study is relevant to the field in investigating this disease, and in some ways addresses a specific gap in the field with the limitations mentioned before, but it is necessary to continue investigating and incorporating sex-specific analyses in the study of SCA1 as these results indicate existence of sex-differences in molecular and cellular pathogenesis as well as in behavioral phenotype in f-ATXN1146Q mice.
We thank the reviewer for recognizing that our manuscript address an important issue in the field.

Round 2
Reviewer 1 Report (New Reviewer)
Comments and Suggestions for Authors
The authors have addressed all the requested revisions, and the manuscript now meets the necessary standards for publication. Therefore, the article is suitable for publication in its current form.
This manuscript is a resubmission of an earlier submission. The following is a list of the peer review reports and author responses from that submission.
Round 1
Reviewer 1 Report
Comments and Suggestions for Authors
The authors investigate differences in sex in terms of behavior assays, pathological assessments and bulk RNAseq of the cerebellum in a novel mouse model for SCA1.
The authors highlighted some of these differences previously in other mouse models of SCA1 in the past [Asher M et al. Cerebellar contribution to the cognitive alterations in SCA1: evidence from mouse models. Hum Mol Genet. 2020 Jan 1;29(1):117-131. doi: 10.1093/hmg/ddz265. PMID: 31696233; PMCID: PMC8216071.]
This current study adds more detail and new assays but doesnt add much to further our understanding into the sex effect in SCA1. The histology needs to be accessible in the paper right now only the bar graphs are presented but the images should be in the paper. The bulk RNAseq dataset was exciting to see but little is done with it other than GO analysis. It would be nice to further explore the sex specific changes and perhaps dive with deeper informatic approaches to find the driving pathways, maybe its linked the estrogen receptor transcription factor?
Overall I found the experimental design to be rigorous and appropriate but information provided to be slightly incremental to what is already known, in other words it reinforces an observation previously made by the authors and others in the field without adding much new insight into the mechanism.
Author Response
We thank the reviewer1 for their helpful comments and suggestions that have improved our manuscript.
Our point to point responses to reviewer’s comments (in italics) are below.
Point 1: “The authors investigate differences in sex in terms of behavior assays, pathological assessments and bulk RNAseq of the cerebellum in a novel mouse model for SCA1. The authors highlighted some of these differences previously in other mouse models of SCA1 in the past [Asher M et al. Cerebellar contribution to the cognitive alterations in SCA1: evidence from mouse models. Hum Mol Genet. 2020 Jan 1;29(1):117-131. doi: 10.1093/hmg/ddz265. PMID: 31696233; PMCID: PMC8216071.] This current study adds more detail and new assays but doesnt add much to further our understanding into the sex effect in SCA1. “
We thank the reviewer for this comment. To clarify how findings described in this manuscript build on our previous findings and increase our understanding of sex-differences in SCA1 we have included paragraphs below in revised introduction and discussion.
Introduction
“Using SCA1 transgenic model, ATXN1[82Q] mice, overexpressing human ATXN1[82Q] 30-60X selectively in Purkinje cells, and SCA1 knock-in model, Atxn1154Q/2Q line, expressing mouse Atxn1[154Q] throughout the brain at endogenous levels, we have previously demonstrated worse motor and cognitive performance in male ATXN1[82Q] mice. In contrast, we found no statistically significant sex-differences in behavioral phenotype in Atxn1154Q/2Q mice. To determine the impact of human vs mouse mutant ATXN1 on sex differences in SCA1 models, here we used a novel SCA1 mouse model, f-ATXN1146Q line13, expressing human ATXN1[146Q] under the endogenous promoter. In addition, to investigate whether sex impacts SCA1 progression, we compared male and female mice at two different stages of disease. Furthermore, to increase mechanistic understanding of sex differences, we analyzed cellular and molecular pathogenesis in female and male SCA1 mice. “
In discussion:
Our previous studies revealed sex differences in cognitive and motor deficits in Purkinje cell specific transgenic SCA1 model, ATXN1[82Q] line, overexpressing human mutant ATXN1 but not in the SCA1 knock-in model, Atxn1154Q/2Q line expressing mouse ATXN1 under endogenous promoter. This raised two important questions: whether these sex differences seen in transgenic model are due to human vs mouse ATXN1 or to exacerbated cerebellar pathology due to ATXN1 30-60 fold overexpression in transgenic SCA1 mice. To address these questions, here we examined sex-differences in the motor and cognitive behavior, in the novel mouse model of SCA1, f-ATXN1146Q mice, expressing human mutant ATXN1 under endogenous promoter. Overall, we found that male f-ATXN1146Q mice are more impacted. Using rotarod and balance beam we found earlier and more severe motor deficits in male f-ATXN1146Q mice at all ages tested. This contrasts with earlier onset of motor deficits in female ATXN1[82Q] mice, but worse deficits in male ATXN1[82Q] mice at later disease stage. Cognitive performance was more complex. Similar to our study in Atxn1154Q/2Q mice, both male and female f-ATXN1146Q mice were similarly impaired in learning and memory in Barnes maze at 12 weeks, indicating the absence of sex-differences in cognitive deficits early in disease. However, at 24 weeks female f-ATXN1146Q mice had worse performance in Bares maze, indicating that with disease progression female SCA1 mice became more cognitively impaired, like human SCA patients.
We found that only male f-ATXN1146Q mice were significantly freezing less in the contextual fear conditioning both during training and testing at 12-13 weeks, indicating the deficits in learned fear response. At 24-25 weeks neither male nor female f-ATXN1146Q mice were impaired compared to their wild-type controls indicating that with aging SCA1 mice normalize freezing in response to noxious stimuli.
Little is understood about failure to gain and weight loss phenotype in SCA1. Surprisingly, while male f-ATXN1146Q mice progressed from failure to gain weight at 12-13 weeks to loss of weight phenotype at 33-35 weeks, female f-ATXN1146Q mice exhibited only failure to gain weight at all ages examined, indicating sex-differences in the weight phenotype in SCA1.
At the level of PC pathology, we did not find any sex-differences, with dendritic atrophy and decrease in calbindin staining, often used to evaluate PC pathology in SCA1 mice, being similar in male and female f-ATXN1146Q. On the other hand, our results show slightly increased astrogliosis and microgliosis in female and male f-ATXN1146Q mice respectively, indicating, for the first time, sex-differences in cellular SCA1 pathogenesis. Thus, it is possible that sex-differences in glial neuroinflammatory phenotypes underly sex-differences in SCA1 behavior.
We also observed differences in molecular SCA1 pathogenesis in SCA1 female and male mice. In general, transcriptional changes were more abundant in female SCA1 mice, indicating for the first time sex-differences in transcriptional changes in SCA. We also found that male f-ATXN1146Q mice had pronounced increase in viral response pathways, consistent with increased microglial density.
Our results indicate potentially worse motor phenotype in male f-ATXN1146Q mice throughout disease, and worse cognitive deficits in female f-ATXN1146Q mice at late disease stage. Studies in patients indicate worse cognitive progression in female SCA patients, consistent with worse cognitive phenotype and increased transcriptional perturbations in female SCA1 mice. On the other hand, we have previously found that sex affects ataxia severity in SCA1 after adjusting for age, disease duration and CAG repeats, with women with SCA1 having SARA scores 3.32 points higher than men with SCA17.
Together these results indicate existence of sex-differences in molecular and cellular pathogenesis as well as in behavioral phenotype in f-ATXN1146Q mice. As we have not seen behavioral sex-differences in Atxn1154Q/2Q mice, it is possible that this is due to expression of human mutant ATXN1 in f-ATXN1146Q mice, or due to the larger number of analyzed mice and or analysis at two different disease stages.
Point 2: The histology needs to be accessible in the paper right now only the bar graphs are presented but the images should be in the paper.
We have included representative images in the revised Figure 3.
Point 3: The bulk RNAseq dataset was exciting to see but little is done with it other than GO analysis. It would be nice to further explore the sex specific changes and perhaps dive with deeper informatic approaches to find the driving pathways, maybe its linked the estrogen receptor transcription factor?
We thank the reviewer for this suggestion. To address driving pathways, we performed additional analysis and in revised Figure 4 we included the analysis of transcriptional factors regulating upregulate and downregulated DEGs in male and female SCA1 mice described in revised results below. Figure 4D is a heatmap of estrogen receptor regulated genes.
To investigate regulatory pathways involved in SCA1 transcriptional changes we performed transcription factor analysis (Figure 4B). Among the three top transcriptional regulators of downregulated DEGs in female f-ATXN1146Q mice was CTCF transcriptional regulator known to interact with estrogen receptor24 and two different Kruppel-like (KLF) factors, well known for their role in disease and inflammation25. Interferon-regulatory factors (IRF) 3 and 9 were the top transcriptional regulators of upregulated DEGs in female f-ATXN1146Q. Wilm’s tumor 1 (WT1), associated with Alzheimer’s disease and synaptic plasticity26 is among the key regulators of downregulated DEGS, while heat shock response factor 4 (HSF 4) a negative regulator of HSFs27 and E24F, involved in Alzheimer’s disease pathigenesis28 are identified as key regulators of upregulated DEGs in male f-ATXN1146Q mice.
We found large and variable perturbations in the expression of estrogen regulated genes in the cerebella of female and male f-ATXN1146Q mice compared to their wild-type controls (Figure 4D, Supplementary Figure 1). Interestingly, certain groups of genes such as Ddx5 Uba5, and Rbfox2 were strongly upregulated in male f-ATXN1146Q mice but decreased in female f-ATXN1146Q mice. Ddx5 encodes DEAD-box RNA helicase DDX5 (also known as p68) inhibitor of anti-viral immune response, and regulator of complement 3 (C3) and myelin basic protein (MBP) expression29,30. Mutations in UBA5 that encodes ubiquitin-like modifier activating enzyme 5 (UBA5) are cause of autosomal recessive spinocerebellar ataxia-24 (SCAR24)31, and splicing regulator Rbfox2 is required for Purkinje cell function32. Estrogen receptor 1 (Esr1) is upregulated in female f-ATXN1146Q mice.

Reviewer 2 Report
Comments and Suggestions for Authors
Review report of manuscript “Sex-differences in a novel mouse model of Spinocerebellar Ataxia Type 1 (SCA1)”
Spinocerebellar Ataxia type 1 (SCA1) is one specific type of Ataxia among a group of inherited diseases of the central nervous system. In SCA1, genetic defects lead to impairment of specific nerve fibers carrying messages to and from the brain resulting in degeneration of the cerebellum. SCA1 is an autosomal dominant disease which means that individuals of either sex are as equally likely to inherit the gene and develop the disease. Each child of a person with SCA1 has a 50 percent chance of inheriting the SCA1 gene. The SCA1 gene passes directly from one generation to the next without skipping generations. The first symptoms are usually incoordination of the hands and trouble with balance when walking.
Dear authors
The manuscript idea is very good and introduced well.
In the present research the authors used different methods for studying the impact of SCA1on behavior of both sex of f-ATXN1146Q mice in comparison to wild-type (WT) control mice. Behavior tests divided into two types: Motor testing and cognitive testing which were the main symptoms appeared in the SCA1 patients. Also, the authors studied immunohistochemistry of brain slices and RNA sequencing.
The results of the study revealed that male f-ATXN1146Q mice are more impacted in all behavior tests. Also, they failure to gain weight at 12-13 weeks and loss weight at 33-35 weeks, while female only failure to gain weight in all ages.
At the level of pathology, the author did not find any differences in PCs but found differences in glial cells.
Transcriptional changes were more abundant in female mice. Consistent with increased microglial density, male f-ATXN1146Q mice had pronounced increase in viral response pathways.
Please add conclusion at the end of the manuscript. Also, add study limitations.
Author Response
We thank the reviewer 2 for their helpful comments and suggestions that have improved our manuscript.
Our point to point responses to reviewer’s comments (in italics) are below.
Point 1:
- Please add conclusion at the end of the manuscript. Also, add study limitations.
In addition to expand discussion included above, we added these limitations and conclusions.
"One limitation of this study is that mouse behavior is not easily interpreted, especially in relation to human behavior. Similarly, SCA1 mice die before the loss of ovarian function, this we cannot compare sex-differences at the age when most female SCA patients are more impaired. Finally, cellular and transcriptional cerebellar species differences between human and mice need to be considered when interpreting our results.
With these limitations, our study significantly increase our understanding of sex-differences in SCA1 mouse models . First, we demonstrate that some sex-differences, such as in motor deficits, can be observed early on and persist with male mice being more impaired. Second, we show that other sex-differences, such as cognitive deficits in Barnes maze are detectable only at late disease stages, with female mice being more impaired. Third, glial cells and not cerebellar Purkinje neurons, show sex-differences in SCA1 mice. Finally, female SCA1 mice show larger number of transcriptional alterations and there are surprising differences in altered pathways and transcriptional factors involved between female and male SCA1 mice. These results also have important implications for preclinical studies in SCA1 mice."
